# TRANSFERABLE AVAILABILITY POISONING ATTACKS

## ABSTRACT

We consider availability data poisoning attacks, where an adversary aims to degrade the overall test accuracy of a machine learning model by crafting small perturbations to its training data. Existing poisoning strategies can achieve the attack goal but assume the victim to employ the same learning method as what the adversary uses to mount the attack. In this paper, we argue that this assumption is strong, since the victim may choose any learning algorithm to train the model as long as it can achieve some targeted performance on clean data. Empirically, we observe a large decrease in the effectiveness of prior poisoning attacks if the victim uses a different learning paradigm to train the model and show marked differences in frequency-level characteristics between perturbations generated with respect to different learners and attack methods. To enhance the attack transferability, we propose *Transferable Poisoning*, which generates high-frequency poisoning perturbations by alternately leveraging the gradient information with two specific algorithms selected from supervised and unsupervised contrastive learning paradigms. Through extensive experiments on benchmark image datasets, we show that our transferable poisoning attack can produce poisoned samples with significantly improved transferability, not only applicable to the two learners used to devise the attack but also for learning algorithms and even paradigms beyond.

## 1 INTRODUCTION

With the growing need for utilizing large amount of data to train machine learning models, especially for training state-of-the-art large-scale models, online data scraping has become a widely used tool. However, the scraped data often come from untrusted third parties, which undesirably empowers adversaries to execute data poisoning attacks more easily. Availability poisoning (Huang et al., 2021; Fowl et al., 2021; Fu et al., 2022; Yu et al., 2022; Segura et al., 2022; He et al., 2023; Zhang et al., 2023), a specific type of data poisoning attacks, has received a lot of attention recently, due to their potential threats to the robustness of model training pipelines. More specifically, an availability poisoning attack aims to reduce the model test performance as much as possible by injecting carefully-crafted imperceptible perturbations into the training data. Existing attack strategies are successful in generating poisoned data samples that are highly effective in lowering the model test accuracy with respect to a given supervised (Huang et al., 2021; Fowl et al., 2021) or unsupervised learner (He et al., 2023; Zhang et al., 2023). However, the effectiveness of these attacks largely relies on the assumption that the victim employs the same learning method to train the model as the reference learner that the adversary uses to devise the attack.

In this paper, we argue that imposing such assumption is unreasonable, as the victim has the flexibility to choose from a wide range of learning algorithms to achieve their objectives, especially considering the rapid advancement of semi-supervised (Chapelle et al., 2006; Berthelot et al., 2019; Sohn et al., 2020; Li et al., 2021) and unsupervised (Chen et al., 2020a;c; Grill et al., 2020; Chen & He, 2021) learning methods. In particular, these approaches can often match or even surpass the performance of supervised learning in various machine learning tasks (Chen et al., 2020b; 2021; Radford et al., 2021; Zheng et al., 2022), which greatly expands the range of options for the victim to train a satisfactory model. The advancement of these alternative methods has prompted us to study the transferability of availability poisoning attacks across different learning methods and paradigms. Unfortunately, we observe a significant decrease on the effectiveness of existing attacks when the victim uses a different learning paradigm to train the model, which renders them less effective in relevant real-world applications (see Figure 1 for a heatmap visualizing the transferability of clean model accuracy and the effectiveness of existing attacks across different victim learners).

We hypothesize that the challenge in transferring poisoning perturbations primarily stems from the differences in attack mechanisms across various learning paradigms. As demonstrated by previous works, for most attacks in supervised learning (Huang et al., 2021; Fu et al., 2022; Yu et al., 2022; Segura et al., 2022), the generated poisoning perturbations exhibit linear separability, providing a shortcut for the model to converge quickly. However, in unsupervised contrastive learning, the poisoning methods work by aligning two augmented views of a single image and are much less linearly separable (He et al., 2023; Ren et al., 2023). By leveraging the insights gained by investigating the transferability of prior availability attacks across a variety of victim learners, we propose a novel method that can generate poisoning samples with significantly improved transferability across different prominent learning methods and paradigms.

**Contributions.** We show the disparate transferability performance of existing availability attacks with respect to different victim learners, spanning across supervised, semi-supervised and unsupervised learning paradigms (Figure 1). To gain a deeper understanding of the marked differences in attack transferability, we visualize poisoning perturbations and discover that linearly separable noises tend to show low-frequency characteristics, whereas less linearly separable noises exhibit an opposite pattern (Figure 2). In particular, we observe that perturbations generated by *Transferable Unlearnable Examples* (Ren et al., 2023) involve both low-freqency and high-frequency components (Figure 2d), aligned with their design idea of enhancing the linear separability of poisoning perturbations generated for unsupervised contrastive learning. However, since these two types of perturbations are quite different in terms of frequency, combining them may compromise the attack effectiveness of each. Fortunately, linear separability is not a necessary property for poisoning to be effective for supervised learners. For instance, *Targeted Adversarial Poisoning* (Fowl et al., 2021) can produce poisoning perturbations with high-frequency features that are not linear separable, yet it achieves state-of-the-art attack performance for supervised learning (Figures 1 and 2b).

Inspired by these observations, we propose *Transferable Poisoning* (TP) to generate poisoning perturbations with shared high-frequency characteristics that can be transferred across different learning paradigms (Section 3). Specifically, we select supervised and unsupervised contrastive learning as our foundational learning paradigms because they are representative, and most other training algorithms are in some way related to them. Built upon on *Targeted Adversarial Poisoning* (Fowl et al., 2021) and *Contrastive Poisoning* (He et al., 2023), our method iteratively leverages information from models trained using supervised and unsupervised contrastive learning to optimize the poisoning perturbations (Section 3.2). By doing so, we ensure that the generated poisons exclusively contain high-frequency features without compromising their efficacy in each individual paradigm. To validate the effectiveness and transferability of our proposed TP, we conduct extensive experiments under various settings. Our results demonstrate that TP is not only highly effective in the two chosen learning paradigms but also exhibits significantly improved transferability to other related training algorithms among different learning paradigms (Sections 4.1 - 4.4). For instance, the test accuracy achieved by the best victim learner trained with the poisoned data produced by our method is as low as $32.60\%$ on CIFAR-10, which indicates an improvement of over $27\%$ in attack effectiveness compared with existing methods (Table 1). We also provide deeper insights to explain the advantages of using iterative optimization, which is much better than a simple post-processing combination or utilizing the information from different learning paradigms together (Section 5).

## 2 RELATED WORK

**Availability Poisoning.** Availability poisoning attacks, also known as indiscriminate data poisoning attacks, aims to undermine the model's overall performance by maliciously manipulating its training data. A line of existing works consider adversaries who attempt to achieve this goal by injecting a small amount of poisoned samples into the clean training dataset (Koh & Liang, 2017; Suya et al., 2021; Lu et al., 2023; Suya et al., 2023). Although their considered threat model does not require direct assess to the original training data, the allowed adversarial modifications to the extra data are usually not restricted to be imperceptible, thus the poisoned data are relatively easier to be distinguished from the normal inputs. In addition, these attacks have been shown effective to poison linear learners, however, they often fail to achieve satisfactory performance when applied to non-convex learning methods such as deep neural networks. Another line of research studies availability poisoning attacks that add imperceptible perturbations to the training data to manipulate

the victim learner (Huang et al., 2021; Fowl et al., 2021; Yu et al., 2022; He et al., 2023; Zhang et al., 2023). In particular, Huang et al. (2021) proposed to craft unlearnable examples based on a reference model with error-minimization perturbations (EM), while Fowl et al. (2021) proposed *Targeted Adversarial Poisoning* (TAP), which generates poisoning perturbations in a reverse way using error-maximization objectives. Several recent works considered to leverage the property of linear separability to create poisoning perturbations more efficiently (Yu et al., 2022; Segura et al., 2022). In addition to supervised learning, the effectiveness of availability poisoning attacks is also validated in unsupervised learning. For instance, Zhang et al. (2023) proposed *Unlearnable Clusters* to address the challenge of agnostic labels, whereas He et al. (2023) focused on studying poisoning attacks for unsupervised contrastive learning methods and proposed *Contrastive Poisoning* (CP).

**Transferable Poisoning.** Different from previous works (Zhu et al., 2019; Aghakhani et al., 2021) which study the transferability on targeted poisoning attacks where the attacking goal is to classify a test image to a specific class, some recent studies explore the transferability of availability poisoning attacks from various perspectives. Huang et al. (2021); Fowl et al. (2021) demonstrate that the poisons generated based on one model architecture can be successfully transferred to another. Similarly, He et al. (2023) consider the scenario where the adversary and victim use different frameworks for unsupervised contrastive learning. *Transferable Unlearnable Examples* (TUE) (Ren et al., 2023) is the closest work to ours, as it also considers transferability in a more generalized manner, encompassing different learning paradigms. They incorporate linear separability into unsupervised unlearnable examples to facilitate transferability between supervised and unsupervised contrastive learning. However, TUE's transferability is limited to the two specific learning paradigms it involves and is challenging to generalize beyond them. In contrast, our method utilizes high-frequency characteristics, a common property shared among effective poisoning across different learning paradigms, to generate perturbations that exhibit much better attack transferability.

# 3 TRANSFERABLE POISONING

## 3.1 PROBLEM SETUP

We consider the setting where there are two parties: an adversary and a victim. The adversary is allowed to add small perturbations to any training data but has no knowledge about how the victim will train the model, including the learning paradigm, model architecture and initialization scheme, and training routine. Once the adversary releases the poisoned version of the training data, no further modification is allowed. The victim only has access to the poisoned data but with the flexibility to choose a training algorithm from various learning paradigms to produce a satisfactory model for the targeted task. The goal of the adversary is to downgrade the test accuracy of the trained model by crafting imperceptible perturbations to training data no matter what learning paradigm the victim adopts. We assume that the learning algorithm considered by the victim has to be competitive in that the performance of its induced model with clean training dataset is good enough.

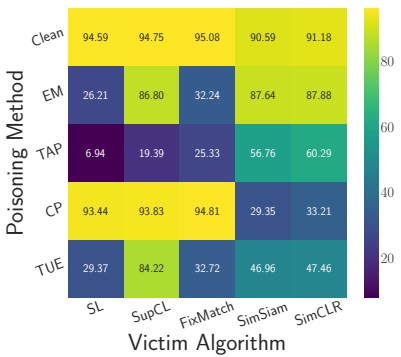

Figure 1: Test accuracy (%) of victim model trained by different algorithms from supervised, semi-supervised and unsupervised learning on clean and various types of poisoned data. Models are trained on ResNet-18 and CIFAR-10.

We focus on the image classification task. Let $[n] = \{1, 2, \ldots, n\}$ and $\mathcal{D}_c = \{(\boldsymbol{x}_i, y_i)\}_{i \in [n]}$ be the clean training dataset. The poisoned dataset is denoted as $\mathcal{D}_p = \{(\boldsymbol{x}_i + \boldsymbol{\delta}_i, y_i)\}_{i \in [n]}$ where $\boldsymbol{\delta}_i$ is the poisoning perturbation added to the $i$-th training input $\boldsymbol{x}_i$. For the ease of presentation, we denote $\mathcal{S}_\delta = \{\boldsymbol{\delta}_i : i \in [n]\}$ as the set of all poisoning perturbations. To ensure the injected noise to be imperceptible to human eyes and difficult for detection, we adopt the commonly imposed $\ell_\infty$-norm constraint $\|\boldsymbol{\delta}\|_\infty \leq \epsilon$ for any $\boldsymbol{\delta} \in \mathcal{S}_\delta$, where $\epsilon > 0$ denotes the predefined poison budget. Following previous works (Fowl et al., 2021; Segura et al., 2022; Ren et al., 2023), we focus our study on sample-wise poisons where the perturbation is generated separately for each sample. Some works (Huang et al., 2021; He et al., 2023) also consider class-wise poisons

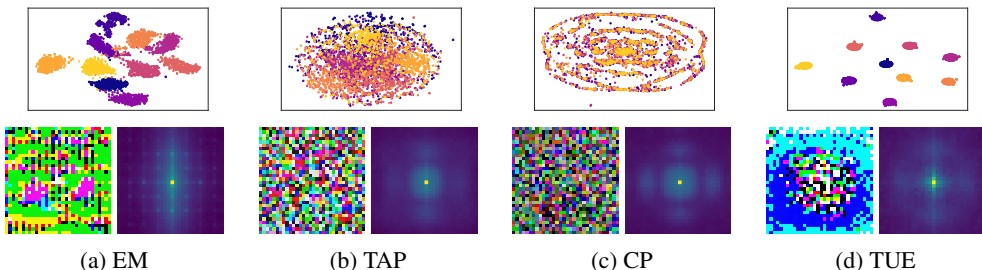

(a) EM        (b) TAP        (c) CP        (d) TUE

Figure 2: Visualizations of the generated noises from EM, TAP, CP and TUE. Top, bottom left and bottom right are t-SNE, original noise and the spectrum respectively.

where all examples in the same class have the same added noise, while it has been shown that this kind of perturbations can be removed by taking average of images in a class (Segura et al., 2022).

## 3.2 METHODOLOGY

**Existing Attacks.** Before presenting our *Transferable Poisoning*, we first study the transferability of existing methods (Figure 1). We test *Unlearnable Examples* (EM) (Huang et al., 2021) and *Targeted Adversarial Poisoning* (TAP) (Fowl et al., 2021) which are targeted for supervised learning, *Contrastive Poisoning* (CP) (He et al., 2023) which is specified for unsupervised contrastive learning, and *Transferable Unlearnable Examples* (TUE) (Ren et al., 2023) which considers the transferability between supervised and unsupervised contrastive learning. We consider that the victim can choose training algorithm from three main learning paradigms, including standard supervised learning (SL), supervised contrastive learning (SupCL) (Khosla et al., 2020), one classical semi-supervised learning method FixMatch (Sohn et al., 2020), and two unsupervised contrastive learning algorithms SimSiam (Chen & He, 2021) and SimCLR (Chen et al., 2020a). In addition, we visualize the generated poisoning perturbations from three aspects to understand the underlying properties (Figure 2). We use t-SNE (van der Maaten & Hinton, 2008) to show the inter-class relation, plot the original noise to visualize the generated patterns and also provide the frequency spectra to analyse the noise characteristics in the frequency domain (see more visualization results in Figure 5 in Appendix B).

According to Figure 1, all of the considered learners show strong performance in producing model with high clean test accuracy, suggesting them as prominent candidates that the victim is likely to pick. In addition, Figure 1 shows that existing availability poisoning approaches have poor transferability across various learning paradigms, which motivates us to propose more transferable attack. As will be shown next, the key idea of our method is to harness the shared properties of poisons that are effective in different learning paradigms. As depicted in Figure 2, linear separability emerges as a crucial characteristic for poisons, such as EM and TUE, when operating in supervised learning. However, this characteristic is not an absolute necessity, as poisons generated by TAP resist linear separation while still achieving state-of-the-art attacking performance. Another key discovery is that linear separability has strong correlation to the frequency of generated perturbations. Interestingly, both TAP and CP, which prove effective in poisoning supervised and unsupervised contrastive learning respectively, exhibit high-frequency features. Building upon those insights, our method aims to generate poisoning perturbations characterized by high frequency. These perturbations are designed to be transferable across various training algorithms within different learning paradigms.

**Transferable Poisoning.** According to the disparate trasferability performance and the visualizations of generated poisoning purterbations with respect to existing availability poisoning attacks, we propose to select supervised and unsupervised contrastive learning as the base learning paradigms for our method, since they are representative and other learning algorithms are in some way related to them. The core idea of our method is to iteratively leverage the information from both selected learning paradigms to search for transferable poisoning perturbations in the constrained perturbation space. Since these two learning paradigms work quite differently, we train two separate models with the same architecture and one for each, while optimizing the poisoning perturbations together. To be more specific, we consider *Targeted Adversarial Poisoning* (TAP) (Fowl et al., 2021) for poisoning the supervised learner. Given a model $\theta_{\text{pre}}$ pretrained by standard supervised learning on clean

---

**Algorithm 1** Transferable Poisoning

---

1: **Input:** clean dataset $\mathcal{D}_c$; pretrained model $\theta_{\mathrm{pre}}$; number of total epochs $T$, number of updates in each epoch $M$; learning rate $\eta$; PGD steps $S_{\mathrm{sl}}$, $S_{\mathrm{cl}}$; attack step size $\alpha_{\mathrm{sl}}$, $\alpha_{\mathrm{cl}}$; poison budget $\epsilon$
2: **for** $t = 1, \ldots, T$ **do**
3:     **for** $m = 1, \ldots, M$ **do**
4:         Sample $\{\boldsymbol{x}_i, y_i\}_{i \in [B]}$ from $\mathcal{D}_c$
5:         **for** $s = 1, \ldots, S_{\mathrm{sl}}$ **do**
6:             Compute $\boldsymbol{g}_i \leftarrow \nabla_{\boldsymbol{\delta}_i} \sum_{i \in [B]} \mathcal{L}_{\mathrm{CE}}(f(\boldsymbol{x}_i + \boldsymbol{\delta}_i; \theta_{\mathrm{pre}}), p(y_i)), \forall i \in [B]$
7:             $\boldsymbol{\delta}_i \leftarrow \Pi_{\mathcal{B}_\epsilon(\mathbf{0})}(\boldsymbol{\delta}_i - \alpha_{\mathrm{sl}} \cdot \mathrm{sgn}(\boldsymbol{g}_i)), \forall i \in [B]$
8:     **for** $m$ in $1, \ldots, M$ **do**
9:         Sample $\{\boldsymbol{x}_i, y_i\}_{i \in [B]}$ from $\mathcal{D}_c$
10:         $\theta \leftarrow \theta - \eta \cdot \nabla_\theta \sum_{i \in [B]} \mathcal{L}_{\mathrm{CL}}(f(\{\boldsymbol{x}_i + \boldsymbol{\delta}_i\}_{i \in [B]}; \theta))$
11:     **for** $m$ in $1, \ldots, M$ **do**
12:         Sample $\{\boldsymbol{x}_i, y_i\}_{i \in [B]}$ from $\mathcal{D}_c$
13:         **for** $s$ in $1, \ldots, S_{\mathrm{cl}}$ **do**
14:             Compute $\boldsymbol{g}_i \leftarrow \nabla_{\boldsymbol{\delta}_i} \sum_{i \in [B]} \mathcal{L}_{\mathrm{CL}}(f(\{\boldsymbol{x}_i + \boldsymbol{\delta}_i\}_{i \in [B]}; \theta)), \forall i \in [B]$
15:             $\boldsymbol{\delta}_i \leftarrow \Pi_{\mathcal{B}_\epsilon(\mathbf{0})}(\boldsymbol{\delta}_i - \alpha_{\mathrm{cl}} \cdot \mathrm{sgn}(\boldsymbol{g}_i)), \forall i \in [B]$
16: **Output:** poisoned dataset $\mathcal{D}_p = \{(\boldsymbol{x}_i + \boldsymbol{\delta}_i, y_i)\}_{i \in [n]}$

---

training dataset $\mathcal{D}_c$, the training objective of TAP is defined as follows:

$$\min_{\mathcal{S}_\delta} \frac{1}{n} \sum_{i \in [n]} \mathcal{L}_{\mathrm{CE}}(f(\boldsymbol{x}_i + \boldsymbol{\delta}_i; \theta_{\mathrm{pre}}), p(y_i)), \text{ s.t. } \|\boldsymbol{\delta}_i\|_\infty \leq \epsilon, \forall i \in [n],$$

where $\mathcal{S}_\delta = \{\boldsymbol{\delta}_i : i \in [n]\}$, $\mathcal{L}_{CE}$ is the cross-entropy loss, $f$ is the neural network, and $p(\cdot)$ denotes some predefined permutation function on the label space. For unsupervised contrastive learning, we adopt *Contrastive Poisoning* (CP) (He et al., 2023), where the model parameter $\theta$ and the set of poisoning perturbations $\mathcal{S}_\delta$ are optimized jointly to solve the following optimization problem:

$$\min_{\theta, \mathcal{S}_\delta} \frac{1}{n} \sum_{j \in [N]} \sum_{i \in [B]} \mathcal{L}_{\mathrm{CL}}(f(\{\boldsymbol{x_i} + \boldsymbol{\delta_i}\}_{i \in [B]}; \theta)), \text{ s.t. } \|\boldsymbol{\delta}_i\|_\infty \leq \epsilon, \forall i \in [n],$$

where $N$ denotes the number of batches, each batch has the same size of $B$ training data with $n = N \cdot B$, and $\mathcal{L}_{\mathrm{CL}}$ denotes the loss used in unsupervised contrastive learning such as InfoNCE. As depicted in Algorithm 1, during the generation process of poisoned data, we iteratively optimize the set of poisoning perturbations $\mathcal{S}_\delta$ based on the pretrained model $\theta_{\mathrm{pre}}$, update the model parameter $\theta$ with unsupervised contrastive learning, and then optimize the perturbations $\mathcal{S}_\delta$ again but with respect to $\theta$. The model parameter is updated via stochastic gradient descent while the poisoning perturbations are optimized by projected gradient descent (PGD) (Madry et al., 2018). The ratio between PGD steps $S_{\mathrm{sl}}$ and $S_{\mathrm{cl}}$ is an essential factor that will influence the attacking performance in each learning paradigm. Figure 1 indicates that poisoning unsupervised contrastive learning is more difficult than poisoning supervised learning, hence the used $S_{\mathrm{cl}}$ is larger than $S_{\mathrm{sl}}$ in default settings. We also discuss the effect of number of PGD steps with more details in Section 4.2.

**Other Alternatives.** To examine the effectiveness of our *Transferable Poisoning* in leveraging the merits of the two selected strategies, *Targeted Adversarial Poisoning* and *Contrastive Poisoning*, we also compare our method with two naive alternative ways for combining these two attacks. One is to directly add the poisons generated by these two approaches separately, while the poisoning budget for each one is set as $\epsilon/2$, denoted as HF, to ensure the final noises is under the budget $\epsilon$. The other is created by first combining the two poisons which are generated under the full poisoning budget $\epsilon$ separately, and then clamping it to meet the constraint, which is denoted as CC. As will be shown in the experiments, our iterative algorithm proves to better leverage the property of poisons working in each learning paradigm and be more effective than using the information from both learning paradigms together, which is evidenced by the markedly improved transferability to other training algorithms. More discussions about the advantage of our method can be found in Section 5.

Table 1: Comparison results of transferability across various learning paradigms and algorithms between different attacking methods. The results are clean accuracy (%) tested on CIFAR-10.

| Attack | SL | SupCL | FixMatch | SimSiam | SimCLR | Best |
|--------|------|-------|----------|---------|--------|-------|
| Clean | 94.59 | 94.75 | 95.08 | 90.59 | 91.18 | 95.08 |
| EM | 26.21 | 86.80 | 32.24 | 87.64 | 87.88 | 87.88 |
| TAP | 6.94 | 19.39 | 25.33 | 56.76 | 60.29 | 60.29 |
| CP | 93.44 | 93.83 | 94.81 | 29.35 | 33.21 | 94.81 |
| TUE | 29.37 | 84.22 | 32.72 | 46.96 | 47.46 | 84.22 |
| HF | 28.47 | 69.05 | 94.99 | 82.54 | 44.04 | 92.13 |
| CC | 10.78 | 57.09 | 94.72 | 56.34 | 40.69 | 94.72 |
| Ours | 8.98 | 30.85 | 32.60 | 17.14 | 28.39 | 32.60 |

Table 2: Evaluation reults of our attack methods on different benchmark image datasets.

| Dataset | Attack | SL | SupCL | FixMatch | SimSiam | SimCLR | Best |
|---------|--------|------|-------|----------|---------|--------|-------|
| CIFAR-10 | Clean | 94.59 | 94.75 | 95.08 | 90.59 | 91.18 | 95.08 |
| | Ours | 8.98 | 30.85 | 32.60 | 17.14 | 28.39 | 32.60 |
| CIFAR-100 | Clean | 75.55 | 73.17 | 69.19 | 62.45 | 63.52 | 75.55 |
| | Ours | 3.46 | 18.36 | 13.38 | 9.72 | 12.26 | 18.36 |
| TinyImageNet | Clean | 55.71 | 56.25 | 53.28 | 41.56 | 42.42 | 56.25 |
| | Ours | 4.58 | 14.43 | 8.03 | 7.14 | 10.11 | 14.43 |

# 4 EXPERIMENTS

**Experimental Settings.** We adopt three commonly used benchmark datasets in image classification task: CIFAR-10, CIFAR-100 (Krizhevsky et al., 2009) and TinyImageNet (Chrabaszcz et al., 2017). The poisoning perturbations are generated by PGD (Madry et al., 2018) on ResNet-18 (He et al., 2016) and SimCLR (Chen et al., 2020a) as the framework of unsupervised contrastive learning (more frameworks are considered in Section 4.3). Following previous works, we set $\epsilon = 8/255$ to constrain the poisoning perturbations (additional results for other perturbation strength are provided in Table 12 in Appendix B). Apart from the five training algorithms adopted in our main experiments, we also consider *Vision Transformer* (ViT) (Dosovitskiy et al., 2021) and *Masked Autoencoders* (MAE) (He et al., 2021) from generative self-supervised learning to further demonstrate the effectiveness of our method (Section 4.3). More detailed experimental settings can be found in Appendix A.

## 4.1 COMPARISONS WITH EXISTING ATTACKS

We first evaluate the transferability of our method on the benchmark CIFAR-10 dataset. As shown in Table 1, existing methods struggle to achieve satisfactory transferability to learning algorithms that were not directly used in the perturbation generation process. More specifically, EM and TAP, originally designed for supervised learning, exhibit poor performance in unsupervised contrastive learning or even supervised contrastive learning (EM). CP primarily impacts the model's performance in unsupervised contrastive learning and has minimal effect on other training algorithms. TUE shows relatively improved transferability, while supervised contrastive learning can largely restore the model's accuracy. As for the two simply combined poisoning perturbations, CC seems to be a better strategy, as it remains effective on standard supervised learning and SimCLR, and can also fool the model trained with supervised contrastive learning and another unsupervised contrastive learning, SimSiam, albeit with more sacrifice, but cannot influence the training with semi-supervised learning algorithm FixMatch. In comparison, our method demonstrates significantly superior transferability across representative learning paradigms. The victim's best testing accuracy drops by at least approximately 30% compared to other methods. We also verify the transferability of our method on CIFAR-100 and TinyImageNet. As demonstrated in Table 2, our approach consistently reduces testing accuracy to a large extent across various learning algorithms and paradigms.

Table 3: Effect of different ratios of PGD steps between $S_{sl}$ and $S_{cl}$ in Algorithm 1.

| Ratio | SL | SupCL | FixMatch | SimSiam | SimCLR |
|---|---|---|---|---|---|
| 1:1 | 7.22 | 19.19 | 23.14 | 41.99 | 39.01 |
| 1:2 | 8.82 | 24.45 | 28.40 | 26.01 | 33.31 |
| 1:3 | 7.48 | 27.41 | 29.69 | 25.50 | 32.49 |
| 1:5 | 8.98 | 30.85 | 32.60 | 17.14 | 28.39 |

Table 4: Effect of different contrastive learning algorithms.

| Framework | SL | SupCL | FixMatch | SimSiam | SimCLR | MoCov2 |
|---|---|---|---|---|---|---|
| SimCLR | 8.98 | 30.85 | 32.60 | 17.14 | 28.39 | 28.05 |
| MoCov2 | 19.59 | 32.86 | 37.35 | 18.29 | 27.60 | 23.92 |

## 4.2 Effect of Iteration Ratios

In Algorithm 1, we have to set two hyperparameters $S_{sl}$ and $S_{cl}$, representing the number of PGD steps to update the perturbations in each iteration with targeted adversarial poisoning and contrastive poisoning respectively. In default settings, we adopt $S_{sl}$ as 1 and $S_{cl}$ as 5 to generate the poisons. We then keep the same $S_{sl}$ and decrease $S_{cl}$ to see the effect of different updating ratios. As demonstrated in Table 3, the generated perturbations are robust to fool supervised learning, showing consistent accuracy when the ratio varies. The influence on supervised contrastive learning and FixMatch has similar patterns, as the poisoning perturbations would be less effective if it is updated with unsupervised contrastive learning for more times. As expected, model trained with SimSiam and SimCLR are affected in the opposite way, more updates with unsupervised contrastive learning makes the poisons more effective. However, we observe that the model's performance is consistently downgraded by our method, showing the stable transferability across various learning paradigms.

## 4.3 More Unsupervised Learning Algorithms

We use SimCLR as the framework of unsupervised contrastive learning to craft the poisoning perturbations, here we involve another quite different framework MoCov2 (Chen et al., 2020c) in both poison generation and evaluation. Table 4 indicates that the results only vary in supervised and semi-supervised learning while the model accuracy learned by other algorithms are quite similar. But still, all the performance can be downgraded by our method to a large extent.

Table 5: Transferability to generative self-supervised learning. ViT is fine-tuned with a linear classifier and MAE is trained from scratch.

| Attack | ViT | MAE |
|---|---|---|
| Clean | 97.36 | 90.00 |
| TAP | 15.46 | 25.79 |
| CP | 96.73 | 88.48 |
| TUE | 20.31 | 21.13 |
| CC | 27.27 | 47.59 |
| Ours | 24.49 | 32.52 |

To further estimate the transferability of our approach, we adopt another type of unsupervised learning, i.e., generative self-supervised learning. Compared to the previous learning algorithms, the difference of this learning paradigm lies in both the model architectures and the training mechanisms. We consider two scenarios which are fine-tuning and training from scratch, and two learners *Vision Transfomer* (ViT) and *Masked Autoencoders* (MAE), with one for each. As Table 5 illustrates, fine-tuning using only a linear classifier (ViT) can yield a well-performed model when the training data is clean. However, attacks can still succeed provided that the poisoning perturbations are effective in supervised learning (Table 1), including TAP, TUE, CC, and our method. This observation holds true for MAE which is trained from scratch. Though updating the entire model can learn more information, the model accuracy will still be heavily downgraded.

## 4.4 Transferability under Model Architectures

In default settings, we use ResNet-18 as the architecture for both supervised and unsupervised contrastive learning to generate the poisoning perturbations, here we test whether such poisons are still

Table 6: Attack transferability from ResNet-18 to other model architectures.

| Architecture | SL | SupCL | FixMatch | SimSiam | SimCLR | Best |
|---|---|---|---|---|---|---|
| ResNet-18 | 8.98 | 30.85 | 32.60 | 17.14 | 28.39 | 32.60 |
| ResNet-34 | 12.85 | 32.52 | 35.86 | 17.11 | 24.97 | 35.86 |
| VGG-19 | 15.85 | 32.58 | 36.68 | 15.13 | 23.66 | 36.68 |
| DenseNet-121 | 10.82 | 35.50 | 22.16 | 12.22 | 24.72 | 35.50 |
| MobileNetV2 | 11.62 | 33.33 | 34.97 | 19.05 | 25.97 | 34.97 |

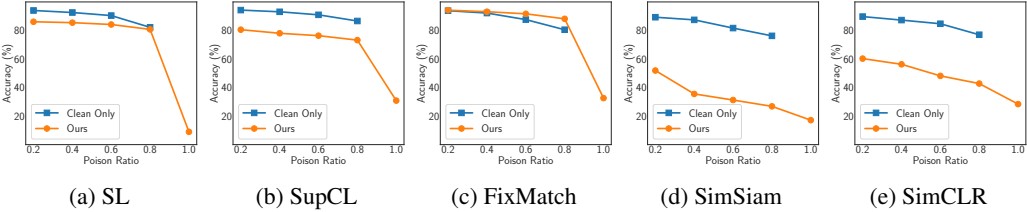

|  (a) SL | (b) SupCL | (c) FixMatch | (d) SimSiam | (e) SimCLR |

Figure 3: Effect of partial poisoning. "Ours" uses the entire training data with the poisoning proportion adjusted, while "Clean Only" merely uses the rest clean training data.

effective if the victim adopts other architectures to learn a model. As can be seen from Table 6, our attack can transfer quite well from ResNet-18 to other five commonly used model architectures including ResNet-34, VGG-19, DenseNet-121 and MobileNetV2, which further demonstrates the strong transferability of our proposed method.

## 4.5 EFFECT OF PARTIAL POISONING

Here, we consider more challenging settings where only partial training data can be poisoned. We report the results in Figure 3. As for the reference where the model is trained with the rest clean data, the testing accuracy shows desirable patterns for all the selected training algorithms as it gradually decreases when the poison ratio increases. In contrast, for supervised, semi-supervised and supervised contrastive learning, the accuracy of the poisoned model will sharply increases when the poison proportion is not 100%, and the model trained with semi-supervised learning even outperforms what trained on clean data. This is understandable since the data is partially poisoned and the algorithm is more likely to involve clean unlabeled data samples during model training, therefore leading to better results. In comparison, poisoning effect on unsupervised contrastive learning is more resilient to the change of poison ratios.

## 5 ANALYSIS

In this section, we provide the insights to explain why our method has better transferability across different learning paradigms compared to other baselines.

**Working Mechanism of Transferable Poisoning.** As depicted in Figures 2 and 4, TUE combines two types of poisoning perturbations which exhibit significantly different frequency characteristics, hence it compromises the effectiveness on each and results in limited transferability. Since our method is based on TAP and CP which both have high-frequency features originally, the generated noises almost have no sacrifice to each (Figure 4). More importantly, as our method iteratively utilizes the information from the models trained with both learning paradigms, the generated noises are not only adaptive to the involved algorithms, but also other related ones. In comparison, other strategies such as forcing the noises to enable one pattern (TUE) or combining different types of noises as a post-processing (CC) can hardly have transferability to other unseen learning paradigms.

**More Evidence.** We conduct two additional experiments to further demonstrate the superiority of our approach. Specifically, we first adopt the poisoning strategy in CP but generate noises with supervised contrastive learning, denoted as *supervised contrastive poisoning* (SCP), and then change

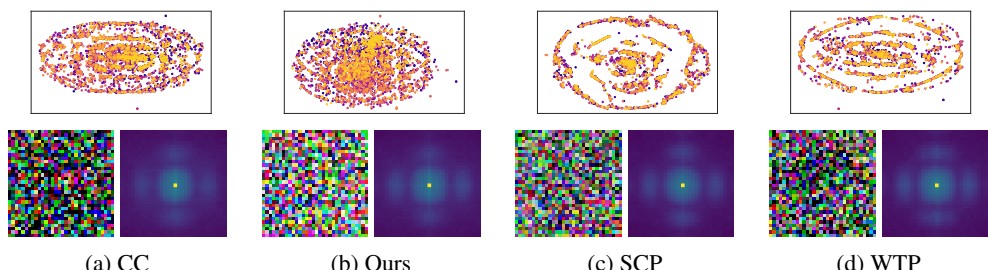

|        (a) CC        |        (b) Ours        |        (c) SCP        |        (d) WTP        |

Figure 4: Visualizations of the generated noises from CC, Ours, SCP and WTP. Top, bottom left and bottom right are t-SNE, original noise and the spectrum respectively.

Table 7: Comparison results of our method in terms of attack effectiveness with SCP and WTP.

| Attack | SL    | SupCL | FixMatch | SimSiam | SimCLR |
|--------|-------|-------|----------|---------|--------|
| Clean  | 94.59 | 94.75 | 95.08    | 90.59   | 91.18  |
| SCP    | 87.42 | 38.76 | 94.57    | 21.06   | 74.91  |
| WTP    | 51.44 | 32.24 | 94.66    | 24.54   | 34.50  |
| Ours   | 8.98  | 30.85 | 32.60    | 17.14   | 28.39  |

the supervised learning part of our method from TAP to EM, denoted as *weak transferable poisoning* (WTP). As illustrated in Table 7, though supervised contrastive learning leverages both label information and techniques from unsupervised contrastive learning, the transferability to these two learning paradigms is quite limited. WTP shows improved transferability compared to SCP, however, both methods cannot be transferred to semi-supervised learning (FixMatch). Comparing the visualization results with respect to different poisoning perturbations, shown in Figures 2 and 4, we can observe that, though SCP and WTP utilize the information from both supervised and contrsative learning, the generated noises are mostly high-frequency and more similar to the noises produced by CP. This similarity explains why they sacrifice the attack performance in supervised learning and other related learning paradigms such as semi-supervised learning.

In summary, the aforementioned experimental results demonstrate the advantage of our method from three perspectives. First, we incorporate information from both learning paradigms during optimization, which is more effective than post-processing combinations like HF and CC. Second, we utilize the property of the availability poisons more reasonably. Our high-frequency perturbation generation mechanism leads to produce high-frequency final perturbations with minimal negative impact, while TUE and WTP combine dissimilar noises, sacrificing transferability on each. Finally, we iteratively update the poisoning perturbations using information from both learning paradigms, which is superior than using them together as in SCP. Supervised contrastive learning combine two learning paradigms but cannot transfer to them. In contrast, our *Transferable Poisoning* is not only effective for the two involved learning paradigms, but also able to transfer to other related ones.

## 6  CONCLUSION

We studied availability poisoning attacks under a more realistic scenario, where the victim has the freedom to choose any learning algorithm to reach the goal of their targeted task. We showed that existing attacks exhibit poor transferability performance in this setting and provided thorough analyses on the features of generated poisoning perturbations to explain this phenomenon. Building on these insights, we introduced *Transferable Poisoning*, which generates high-frequency poisoning perturbations through iterative utilization of information from different learning paradigms. Extensive experiments consistently demonstrate that our method exhibits superior transferability across diverse learning paradigms and algorithms compared to prior attacking methods. Our research uncovers that availability poisoning attacks are more severe than previously anticipated, and we hope it can inspire the community to delve deeper into this field in the future.

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

# A  DETAILED EXPERIMENTAL SETTINGS

## A.1  DATASETS DESCRIPTION

In our experiments, we adopt three commonly used benchmark datasets for classification tasks, i.e., CIFAR-10, CIFAR-100 and TinyImageNet. Among them, CIFAR-10 has 60000 images for 10 classes and each class has the same number of samples; CIFAR-100 is also a balanced datasets which has 100 classes with 600 number of images for each class, both CIFAR-10 and CIFAR-100 has the size of $32 \times 32 \times 3$ for each image. TinyImageNet has 200 classes, and each class has 500 training samples and 50 testing samples, the size for each data sample is $64 \times 64 \times 3$.

## A.2  LEARNING PARADIGMS DESCRIPTION

In our experiments, we focus on three main learning paradigms, supervised, semi-supervised and un-supervised learning for training a machine learning model. As for supervised learning, we consider the standard one which directly utilizes the ground truth labels, and also involve the algorithm which applies contrastive learning in supervised setting, i.e., Supervised Contrastive Learning. As for semi-supervised learning, we adopt FixMatch, which is a combination of two classical approaches (consistency regularization and pseudo-labeling) in semi-supervised learning and achieves state-of-the-art performance. As for unsupervised learning, we focus self-supervised learning which attains comparative and even better performance as supervised and semi-supervised learning. It comprises generative models and unsupervised contrastive learning where the former learns representations with autoencoding and masked image encoding while the latter pulls together the anchor image and a positive sample and push apart the anchor and the other negative samples in feature space. We select SimCLR, MoCov2 and SimSiam for unsupervised contrastive learning, and Vision Transformer and Masked Autoencoders for generative models, which are representative for these two types of self-supervised learning.

## A.3  POISON GENERATION

We train model $\theta_{\text{pre}}$ for 40 epochs with SGD optimizer with momentum 0.9 and weight decay $5 \times 10^{-4}$, and initial learning rate of 0.1 which is decayed by a factor of 0.1 at 15-th, 25-th and 35-th epoch. The iterative training epochs $T = 250$ and the number of updates in each epoch $M = |\mathcal{D}_c|/B$ where $|\mathcal{D}_c|$ is the size of the clean dataset and $B$ is the batch size, which means we pass all the training data samples in each epoch. The model $\theta$ is updated with SGD optimizer with momentum 0.9 and weight decay $1 \times 10^{-4}$, and learning rate $\eta = 0.5$ for SimCLR as the framework. The poisoning perturbations $\boldsymbol{\delta}$ is updated based on model $\theta_{\text{pre}}$ with number of PGD steps $S_{\text{sl}} = 1$ and step size $\alpha_{\text{sl}} = 0.05 \times \epsilon$, and the permutation $p$ is naive as $p(y) = (y + 1)\%C$, where $C$ is the number of classes for each dataset. As for the update based on model $\theta$, the number of PGD steps is 5 for CIFAR-10 and 1 for CIFAR-100 and TinyImageNet, and the step size $\alpha_{\text{cl}} = 0.05 \times \epsilon$ as well. The poison budget $\epsilon = 8/255$.

The cross-entropy loss for supervised learning for each example $(\boldsymbol{x}, y)$ is defined as follows:

$$\mathcal{L}_{\text{CE}}(f(\boldsymbol{x}; \theta), y) = -\sum_{i=1}^{C} y_i \log \left[ f(\boldsymbol{x}; \theta) \right]_i,$$

where $C$ is the total number of classes. $y_i$ equals 1 only if the sample belongs to class $i$ and otherwise 0, and $[f(\boldsymbol{x}; \theta)]_i$ is the $i$-th element of the prediction posteriors. And the InfoNCE loss is:

$$\mathcal{L}_{\text{InfoNCE}}(f(\{\boldsymbol{x_i}\}_{i \in [B]}; \theta)) = -\log \frac{\exp\left(\text{sim}(f(\boldsymbol{x_i}; \theta), f(\boldsymbol{x'_i}; \theta))/\tau\right)}{\sum_{k=1}^{2B} \mathbb{1}(k \neq i) \exp\left(\text{sim}(f(\boldsymbol{x_i}; \theta), f(\boldsymbol{x_k}; \theta))/\tau\right)},$$

where $\boldsymbol{x}'_i$ is another augmented version of $\boldsymbol{x}_i$, $\text{sim}(\cdot, \cdot)$ denotes some similarity function, $\mathbb{1}(k \neq i)$ is an indicator function with value 1 when $k \neq i$, and $\tau$ is a temperature hyperparameter. And this loss is also termed as *NT-Xent* in the paper of SimCLR.

Table 8: Comparison results of transferability across various learning paradigms and algorithms between different attacking methods. The results are clean accuracy (%) tested on CIFAR-100.

| Attack | SL | SupCL | FixMatch | SimSiam | SimCLR | Best |
|--------|------|-------|----------|---------|--------|-------|
| Clean | 75.55 | 73.17 | 69.19 | 62.45 | 63.52 | 75.55 |
| EM | 15.57 | 69.07 | 36.53 | 52.81 | 55.26 | 69.07 |
| TAP | 3.56 | 21.15 | 13.59 | 27.03 | 30.69 | 30.69 |
| CP | 72.34 | 66.70 | 67.18 | 10.53 | 12.65 | 67.18 |
| TUE | 1.34 | 68.54 | 22.32 | 53.04 | 55.41 | 68.54 |
| Ours | 3.46 | 18.36 | 13.38 | 9.72 | 12.26 | 18.36 |

Table 9: Defending against TP using JPEG. The results are clean accuracy (%) tested on CIFAR-10.

| Attack | SL | SupCL | FixMatch | SimSiam | SimCLR | Best |
|--------|------|-------|----------|---------|--------|-------|
| Clean | 94.59 | 94.75 | 95.08 | 90.59 | 91.18 | 95.08 |
| No defense | 8.98 | 30.85 | 32.60 | 17.14 | 28.39 | 32.60 |
| After JPEG | 51.83 | 67.05 | 73.82 | 49.26 | 50.41 | 73.82 |

## A.4 EVALUATION

For supervised learning, we train the model for 100 epochs with SGD optimizer with momentum 0.9 and weight decay $5 \times 10^{-4}$, and initial learning rate of 0.1 which is decayed by a factor of 0.1 at 60-th, 75-th and 90-th epoch. For semi-supervised learning (FixMatch), we use the same hyper-parameters in the original paper, i.e., weight of unlabeled loss $\lambda_u = 1$, the relative size of labeled and unlabeled data $u = 7$, the threshold for pseudo labels $\tau = 0.95$, the model is trained for 200 epochs with SGD optimizer with momentum 0.9 and weight decay $5 \times 10^{-4}$, and initial learning rate of 0.03 which is decayed by a cosine scheduler, the number of labeled data samples are 4000, 10000 and 20000 for CIFAR-10, CIFAR-100 and TinyImageNet respectively. For supervised contrastive learning, SimSiam and SimCLR, we use the same setting for CIFAR-10 and CIFAR-100, i.e., we first pretrain the model for 1000 epochs, the otimizer is SGD with momentum 0.9 and weight decay $1 \times 10^{-4}$, the initial learning rate is 0.5 which is decayed by a cosine scheduler, the temperature for the loss in supervised contrastive learning and SimCLR is 0.5. We then train the linear classifier for 100 epochs with SGD optimizer with momentum 0.9 and weight decay 0, and initial learning rate of 1.0 which is decayed by a factor of 0.2 at 60-th, 75-th and 90-th epoch. For TinyImageNet, we pretrain the model for 1000 epochs, the otimizer is SGD with momentum 0.9 and weight decay $1 \times 10^{-6}$, the initial learning rate is 0.15 which is decayed by a CosineAnnealingLR scheduler, the temperature for the loss in supervised contrastive learning and SimCLR is 0.5, and the hyper-parameters for linear probing is the same as for CIFAR-10 and CIFAR-100.

## B ADDITIONAL RESULTS

Table 10: The time (h) for generating the poisoning perturbations of CIFAR-10.

| Attack | EM | TAP | CP | TUE | Ours |
|--------|------|------|-------|------|-------|
| Time | 0.21 | 0.63 | 44.67 | 2.15 | 15.61 |

Table 11: Effect of different contrastive learning algorithms for TP and baseline methods.

| Attack | Framework | SL | SupCL | FixMatch | SimSiam | SimCLR | MoCov2 |
|--------|-----------|-------|-------|----------|---------|--------|--------|
| CP | SimCLR | 93.44 | 93.83 | 94.81 | 29.35 | 33.21 | 77.87 |
|    | MoCov2 | 93.38 | 94.28 | 92.87 | 51.53 | 45.62 | 43.09 |
| TUE | SimCLR | 29.37 | 84.22 | 32.72 | 46.96 | 47.46 | 82.37 |
|     | MoCov2 | 11.18 | 87.61 | 52.89 | 75.73 | 76.31 | 78.49 |
| Ours | SimCLR | 8.98 | 30.85 | 32.60 | 17.14 | 28.39 | 28.05 |
|      | MoCov2 | 19.59 | 32.86 | 37.35 | 18.29 | 27.60 | 23.92 |

Table 12: Attack performance of our method under different poisoning budget.

| Epsilon | SL | SupCL | FixMatch | SimSiam | SimCLR |
|---------|-------|-------|----------|---------|--------|
| 0 | 94.59 | 94.75 | 95.08 | 90.59 | 91.18 |
| 4/255 | 31.84 | 61.07 | 62.08 | 59.90 | 51.87 |
| 8/255 | 8.98 | 30.85 | 32.60 | 17.14 | 28.39 |
| 16/255 | 6.77 | 28.10 | 20.62 | 22.96 | 26.73 |

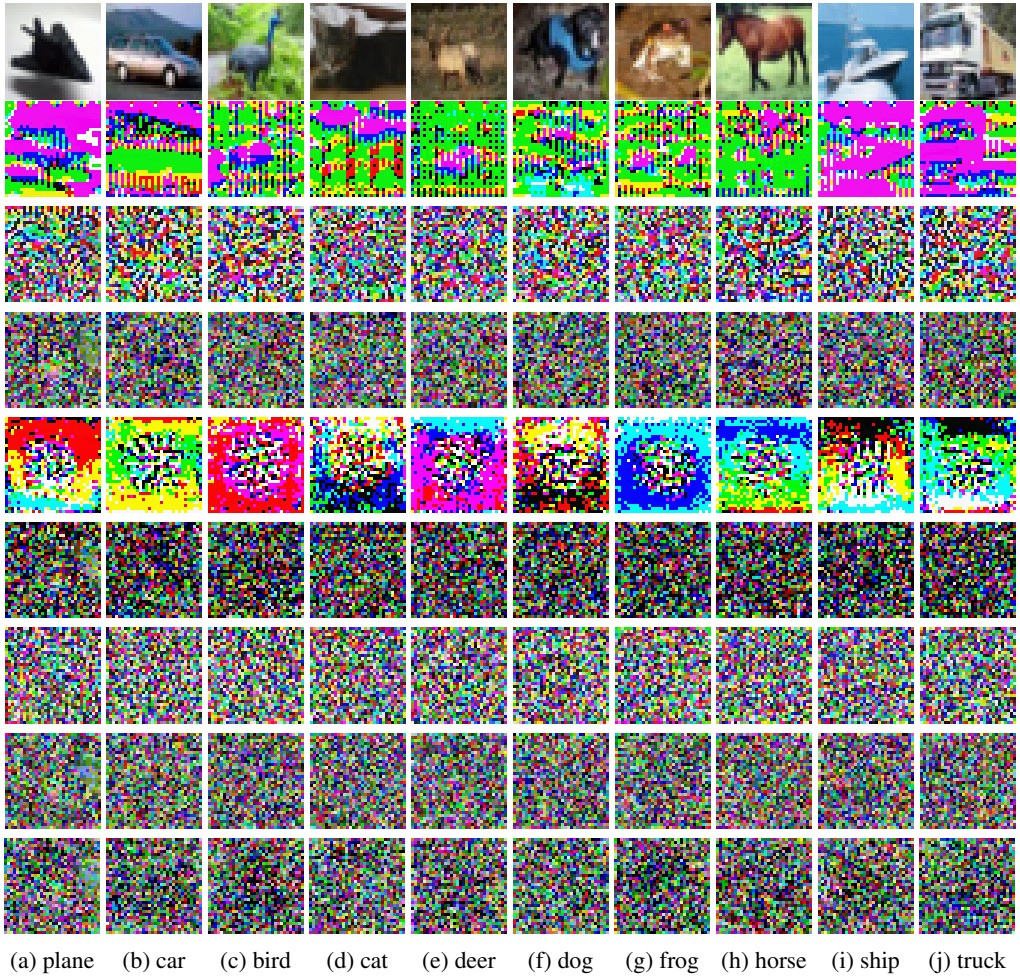

(a) plane  (b) car  (c) bird  (d) cat  (e) deer  (f) dog  (g) frog  (h) horse  (i) ship  (j) truck

Figure 5: Visualizations of perturbations for CIFAR-10 images with one for each class. From top to bottom are the original images, noises generated by EM, TAP, CP, TUE, CC, Ours, SCP and WTP.

