# OpenReview forum: "Transferable Availability Poisoning Attacks"
_ICLR.cc/2024/Conference — Submitted to ICLR 2024_

### Official Review · Reviewer_f8DL · 2023-10-30

**Soundness:** 1 poor
**Presentation:** 3 good
**Contribution:** 2 fair
**Rating:** 3
**Confidence:** 4

**Summary:**

This paper proposes to iteratively update poisons using TAP and CP and the resulting poisoning attack transfers across SL, CL, Semi-Supervised Learning.

**Strengths:**

1. The motivation is convincing that availability poisoning attack should be effective for different learning paradigms.
2. The proposed algorithm improves the transferability of poisoning attack across evaluation algorithms including SL, CL, Semi-Supervised Learning on CIFAR-10, CIFAR-100 and Tiny-ImageNet.
3. This paper provides plenty of investigation on alternative methods including HF, CC, SCP, and WTP.

**Weaknesses:**

1. The authors should check their code regarding the CL and SupCL evaluation for poisoning attacks, especially how transforms (augmentations) are performed when preparing the poisoned dataset. It seems that the data augmentation for each image is fixed before the CL (SupCL) training starts in their code. For example, in this case, each positive pair consists of two fixed augmented images. However, the CL (SupCL) algorithm applies different data augmentations each time it reads an image. Once this part of code is not properly configured, the reliability of the algorithm is questionable. For these reasons, I have significant concerns about the CL and SupCL evaluation results reported in this paper.
2. The novelty of the proposed method is limited. The core of the TP algorithm is a just combination of two existing algorithms, i.e. iteratively updating poisons using TAP and CP to inherit their advantages.
3. The paper seems inaccurately describe the existing work. “Targeted Adversarial Poisoning (Fowl et al., 2021) can produce poisoning perturbations with high-frequency features that are not linear separable, ...” However, as discussed in Table 1 of [a], TAP perturbations are linear separable.
4. The main machanism of TP is kind of vague. This paper claims that “we propose Transferable Poisoning (TP) to generate poisoning perturbations with shared high-frequency characteristics that can be transferred across different learning paradigms (Section 3).” It is unclear why high-frequency perturbation should be a goal for generating transferable poisoning attacks. The proposed algorithm even does not explicitly optimize the frequency of perturbations. In Figure 4, all four attacks are high-frequency but three of them, namely CC, SCP, and WTP have poor transferability. Does it contradict the claim?
5. Besides the first concern about CL evaluation, I have also concerns about the comparison of attack performance.
(1)	Table 1,2,4 do not consider BYOL that was used in the paper of CP. What is the attack performance of TP against BYOL on CIFAR-10 and CIFAR-100?
(2)	In Table 1, is SimCLR used for CP and TUE perturbation generation? However, SimCLR is not surely the best generation algorithm for transferability of CP and TUE. To be a fair comparison, it is helpful to consider CP and TUE attacks based on other CL algorithms as well as class-wise CP attacks which are effective for SL.
(3)	It would be better to compare your TP with existing powerful poisoning methods (UE, AP, CP and TUE) on larger datasets like CIFAR-100 and TinyImageNet.
6. As the algorithm involves contrastive training, it possibly costs too much time to generate TP perturbations, especially on Tiny-ImageNet and ImageNet. Can you provide time consuming comparison of your TP with existing transferable methods like AP, (class-wise) CP, and TUE? Considering availability poisoning attacks are a sort of data protection means, low efficiency of generation might hinder the applications in real world scenarios.

[a] Da Yu, Huishuai Zhang, Wei Chen, Jian Yin, and Tie-Yan Liu. Availability attacks create shortcuts.

Clarity, Quality, Novelty And Reproducibility:
I hope the author would make more clarificaitons about the main machecism of TP and descriptions about existing work. The novelty of proposed algorithm is hindered by the combination of two existing algorithms. For reproducibility, the first concern about CL and SupCL evaluation is vital.

**Questions:**

See Weakness.

---

> ### Author Response · Authors · 2023-11-20
>
> 1. Code issue for evaluating CL and SupCL?
>
> For evaluating CL and SupCL, we follow the implementation from CP[3] and Supervised Contrastive Learning[5] to perform data augmentations, that is, we treat the poisoned data samples as clean ones and apply data augmentations on them. All the data augmentations are in a random manner, so I don’t think the augmented images will be fined each time. You can also check the code of these two papers for how the evaluation of poisoning attacks is performed.
>
> 2. The poisoning method is not novel?
>
> Please refer to our answer to Q2 in global response.
>
> 3. Inaccurate description of previous works?
>
> It is true that TAP perturbations are also linearly separable, but they are less linearly separable than the perturbations from EM and TUE. We will revise this part in the later version. More importantly, what we want to emphasize is the difference of these poisons on the frequency level, and we leverage TAP and CP which share similar high-frequency characteristics.
>
> 4. Why does high frequency help transferability?
>
> Please refer to our answer to Q3 in global response. The poisons generated by TAP (specified for supervised learning) and CP (specified for unsupervised contrastive learning) share similar high-frequency patterns. Since TP is based on these two methods, the generated poisons will also have high-frequency characteristics, and the poisons will not only be effective in supervised and unsupervised contrastive learning, but can also transfer to other related learning paradigms. Other alternatives (CC, SCP and WTP) that also have high-frequency patterns but have poor transferability, which further demonstrates that the way to utilize the information from supervised and unsupervised contrastive learning is very important.
>
> 5. More results?
>
> First, we provide the results for other baseline methods in Table 8 in the updated paper, and it shows that our method still performs better than the baseline methods. We also change the framework of contrastive learning for other methods (CP and TUE) and report the results in Table 11 in the updated paper. Our method demonstrates more superiority in this setting.
>
>
> 6. The computational cost?
>
> Please refer to the new Table 10 in the updated paper. As our method involves contrastive learning, the optimization time is comparatively longer, but it is still less than CP.

---

> > ### Comment · Reviewer_f8DL · 2023-11-22
> >
> > Upon checking the repositories of SupCL, CP, and this paper again, I found that your code for data preparation differs significantly from CP and SupCL. Specifically, in the construction of poisoned datasets such as `TPCIFAR10`, you perform augmentations in the `__init__` method. This implies that the augmentations applied to each image are fixed initially, and the evaluation algorithms utilize these fixed views throughout the subsequent training. On the other hand, SupCL and CP perform augmentations in the `__getitem__` method, resulting in different augmentations being applied to each image in different epochs. I think this essential difference will affect the results of the SupCL. SimCLR, SimSiam, and MoCo v2 the authors reported in this paper.  I will keep my rating.

---

### Official Review · Reviewer_miSr · 2023-10-31

**Soundness:** 3 good
**Presentation:** 3 good
**Contribution:** 2 fair
**Rating:** 5
**Confidence:** 3

**Summary:**

This paper tackles the challenge of availability poisoning attacks in machine learning, where training data is manipulated to degrade model performance. The authors highlight the limited effectiveness of existing attacks when the victim uses a different learning paradigm and introduce Transferable Poisoning (TP), a novel method enhancing attack transferability across various learning algorithms. TP generates high-frequency poisoning perturbations using both supervised and unsupervised contrastive learning paradigms. Extensive experiments on benchmark datasets demonstrate TP's superior performance in ensuring attack effectiveness, regardless of the victim's learning approach.

**Strengths:**

The paper introduces Transferable Poisoning (TP) which significantly enhances the transferability of availability poisoning attacks across various learning algorithms and paradigms. This addresses a gap in existing poisoning strategies, which often assume the victim will use the same learning method as the adversary.
The authors provide extensive experimental results on benchmark image datasets, demonstrating that TP outperforms existing methods in terms of attack transferability. This thorough validation strengthens the credibility of the proposed method.

**Weaknesses:**

While TP introduces improvements in transferability, the contribution appears to be incremental. The method essentially combines supervised and unsupervised contrastive learning paradigms to generate high-frequency poisoning perturbations. However, this combination does not seem to bring a substantial novelty or a paradigm shift for poisoning attacks.

The paper could be strengthened by providing a more solid theoretical foundation for why and how TP improves transferability across learning paradigms. The current explanation relies heavily on empirical observations, which, while valuable, do not provide a comprehensive understanding of the underlying phenomena.

**Questions:**

Is there a theoretical basis that explains why combining supervised and unsupervised contrastive learning paradigms enhances the transferability of poisoning attacks?

Besides test accuracy degradation, what other metrics (such as robustness, perceptibility of perturbations, and computational efficiency) have been considered to evaluate the effectiveness of TP?



--after rebuttal--
Thanks for the response. My concerns about novelty remain. Thus, I would maintain my score.

---

> ### Author Response · Authors · 2023-11-20
>
> 1. The poisoning method is not novel?
>
> Please refer to our answer to Q2 in global response.
>
> 2. Theoretical analysis on why and how TP provides transferability
>
> Intuitively, as TP utilizes the similar high-frequency patterns in the poisons generated by TAP and CP which are effective for supervised and unsupervised contrastive learning respectively, the finally generated poisons can be effective for these two training algorithms and other related ones such as supervised contrastive learning and semi-supervised learning. We will try to provide more theoretical analysis later.
>
> 3. Other evaluation metrics for TP?
>
> We provide the computational cost comparison with respect to different poisoning methods in the new Table 10 in the updated paper. As our method involves contrastive learning, the optimization time is comparatively longer, but it is still less than CP.

---

### Official Review · Reviewer_uUSA · 2023-11-01

**Soundness:** 3 good
**Presentation:** 3 good
**Contribution:** 3 good
**Rating:** 6
**Confidence:** 3

**Summary:**

This paper aims to study availability based data poisoning attacks, i.e., the goal of the attacker is to degrade the overall accuracy of a machine learning model trained on a poisoned training dataset. Existing poisoning methods assume the attacker knows the learning paradigm used for training the model, but this assumption is often unrealistic. This work proposed a transferable poisoning attack that crafts perturbations to the dataset that can be transferred to different supervised/unsupervised learning methods. Experimental results show that this poisoning attack outperforms existing poisoning attacks when transferred to almost all learning paradigms.

**Strengths:**

The topic of transferable poisoning attacks is very interesting. In the real world, an attacker is unable to know the learning method that would be used to train the model.

Experiments have been done for SOTA learning paradigms in recent years.

**Weaknesses:**

This attack (and all previous attacks) needs to poison almost 100% of the dataset to perform well, which is unrealistic in practice. Although this setup is aligned with previous works.

An inconsistency in writing. Section 3.2 mentioned “our method aims to generate poisoning perturbations characterized by high frequency.” But this logic is not reflected in the method design.

The experiment is not very systematic. Baseline methods are only compared on the CIFAR-10 dataset. In Figure 3, poisoning ratio = 0 (clean accuracy) is not shown.

This method may be computationally hard to be applied to 224*224 images, e.g., ImageNet Images. But I think this problem also exists in previous works.

The defenses (e.g., empirical defenses and provable defenses) against poisoning attacks are not discussed. Also, the defenses are not considered.

The proposed method is a combination of two existing methods. The technique contribution is not strong.

**Questions:**

See above.

---

> ### Author Response · Authors · 2023-11-20
>
> 1. The setting that poisoning 100% training data is not practical?
>
> Please refer to our answer to Q1 in global response.
>
> 2. The high frequency is not reflected in method design?
>
> Please refer to our answer to Q3 in global response.
>
> 3. Baseline methods for other datasets?
>
> Please refer to Table 8 in our updated paper for the results tested on CIFAR-100. Our method still performs better than baseline methods.
>
> 4. High computational cost for large-scale dataset?
>
> As our method needs to optimize the poisons with unsupervised contrastive learning, we admit that the computational cost is comparatively high. But like what the reviewer commented, the scalability issue also exists in previous works. Improving the efficiency and scalability of poisoning attacks is an interesting future work.
>
> 5. No defense is discussed?
>
> Liu et al.[6] propose to use image compression operation to defend against availability poisoning attack. They claim that a simple JPEG operation is effective to defend against poisons with high frequency and requires much less computational cost than adversarial training. Thus we apply JPEG to defend against our method and report the results in table 9 in the updated paper. It shows that the JPEG operation can indeed recover the accuracy to some extent, while they are still lower than the model trained with clean data.
>
> 6. The poisoning method is not novel?
>
> Please refer to our answer to Q2 in global response.

---

### Official Review · Reviewer_TTnu · 2023-11-02

**Soundness:** 2 fair
**Presentation:** 2 fair
**Contribution:** 2 fair
**Rating:** 3
**Confidence:** 4

**Summary:**

This paper studies the transferability of indiscriminate poisoning attacks. The paper first shows that the transferability of pre-computed poisoning samples is low across different learning algorithms. To increase the transferability of such poisoning samples, the paper proposes "transferable poisoning," an algorithm to improve the transferability. The key idea is to consider both supervised and unsupervised learning algorithms simultaneously in crafting, such that both latent representations and logits are optimized to increase the target model's loss during training. The paper runs experiments with three image classification benchmarks and shows that the attack is more transferable than existing baselines.

**Strengths:**

1. The paper proposes an indiscriminate attack with increased transferability.
2. The paper show the attack's effectiveness empirically.

**Weaknesses:**

1. The attack is non-practical; it needs the training data to be compromised completely.
2. The attack (when not 100% training samples are compromised) is not effective in supervised learning algorithms.
3. The frequency analyses are not scientifically rigorous.
4. The novelty of this new poisoning attack is weak.
5. No defense was discussed.

Detailed comments:


[Poisoning 100% of the Training Data Is Non-Practical]

I am confident that the paper studies a non-practical scenario: an adversary who poisons 100% of the training data to degrade the accuracy; even with 80% of the training data being compromised, the supervised model retains most of its accuracy compared to clean models.

I haven't seen any real-world scenarios where one allows an adversary to poison 100% of the training data. In any case, the training starts at least with the dataset containing 50% of clean samples.

Prior work on poisoning defenses also theoretically showed that if an adversary can compromise 50% of the training data, there's no guarantee that learning will be empirically successful.

For these reasons, I also don't think that the transferability should not be measured with the completely compromised datasets.


[Not Effective against SL, SupCL, FixMatch]

My takeaway is that even with the datasets that contain 80--100% of poisoning samples, the three supervised learning algorithms can retain the original accuracy.

This means (1) the poisoning attack, even with the increased transferability, is weak and easy to defeat, or (2) the three algorithms are designed to be robust inherently to the distributional shifts.

However, for this paper, this observation is a weakness as the stronger attack is not actually strong in some settings where a victim does not employ any defenses; the victim could trivially depend on the attack.


[Frequency Analyses Are Not Rigorous]

I don't think the paper's claims about the high-frequency components are not scientifically backed by the results. "High" means that there will be a quantifiable property, and when used, the proposed poisoning attacks show large numbers. However, the paper is not.

The paper also claims that the high-frequency characteristics are "different." But it is also not a property that the paper scientifically measures (or compares with criteria). The results are only drawn from the visual analysis.

It is particularly important to make this claim scientifically rigorous as the reason this poisoning attack is transferable is the paper claims that the perturbations of the attack are the high-frequency ones.


[Incremental; Novelty Over the Prior Work]

It is less surprising that combining the classification- and representation-level losses can lead to general-purpose poisoning samples. Similar techniques have been studied, e.g., one considers all the layer outputs in a neural network for adversarial-example crafting, ensembling multiple models, unrolling the training steps to synthesize effective, transferable targeted poisoning attacks iteratively, etc.


[No Defense Evaluation]

Since the attack requires data with 100% poisoning samples, I believe defeating this poisoning attack (or breaking the transferability) is straightforward. I also want to see the discussion about them as a part of responsible vulnerability disclosure.


Overall, for those reasons, I am leaning toward rejection.

**Questions:**

My questions are in the detailed comments in the weakness section.

**Details Of Ethics Concerns:**

No concern about the ethics

---

> ### Author Response · Authors · 2023-11-20
>
> 1. The setting that poisoning 100% training data is not practical?
>
> Please refer to our answer to Q1 in global response.
>
> 2. The method is not effective against SL, SupCL and FixMatch?
>
> The main setting of our method is to poison 100% training data, and our method is effective against all the selected learning paradigms in this setting. We admit that when the poisoning ratio is not 100%, the attack is not quite effective against SL, SupCL and FixMatch, and this phenomenon is also reported by previous works[1,2] which are specified for supervised learning. We plan to explore the poisoning attack transferability under partial-poisoning settings in future work.
>
> 3. Provide more rigorous analysis about frequency?
>
> Please refer to our answer to Q3 in global response.
>
> 4. The poisoning method is not novel?
>
> Please refer to our answer to Q2 in global response.
>
> 5. No defense is discussed?
>
> Liu et al.[6] propose to use image compression operation to defend against availability poisoning attack. They claim that a simple JPEG operation is effective to defend against poisons with high frequency and requires much less computational cost than adversarial training. Thus we apply JPEG to defend against our method and report the results in table 9 in the updated paper. It shows that the JPEG operation can indeed recover the accuracy to some extent, but they are still lower than the model trained with clean data.

---

### Official Review · Reviewer_s6uj · 2023-11-06

**Soundness:** 2 fair
**Presentation:** 3 good
**Contribution:** 2 fair
**Rating:** 5
**Confidence:** 4

**Summary:**

The authors address the problem of generating indiscriminate poisoning attacks that can be applied across various learning paradigms, including supervised, semi-supervised, and unsupervised learning. First, they assess the effectiveness of several existing strategies in terms of their transferability. Then, the authors introduce a novel approach called "Transferable Poisoning", which combines two strategies to achieve strong transferability across the considered learning paradigms.

**Strengths:**

The paper provides an interesting contribution to a significant topic, the transferability of poisoning attacks across diverse learning paradigms, an area that has seen limited exploration. The authors introduce a straightforward and intuitive yet effective method, which is tested on a broad range of experiments. The manuscript is well-written overall, though there is space for improvement.

**Weaknesses:**

I believe this work has two main weaknesses:

1. There is little emphasis on the motivation for studying transferability across learning paradigms, which limits the impact of the findings. It would greatly benefit the paper to explain the practical contexts where this kind of transferability might be relevant, offering concrete examples and detailing how the conducted experiments address these cases. This is particularly crucial given the pragmatic nature of the contribution.

2. While the frequency analysis provides an intriguing perspective on the problem, it could be explored more thoroughly. The authors argue that generating high-frequency perturbations is key to crafting transferable poisoning attacks. This idea is supported by a simple frequency analysis of perturbations generated by existing methods, and so the proposed approach combines two existing ‘high-frequency’ poisoning schemes. Nevertheless, there is no attempt to substantiate this claim with evidence confirming the necessity of high-frequency perturbations for achieving transferability.

In addition to the above points, I have the following minor concerns:

1. More details should be provided on the considered poisoning methods, the associated learning paradigms, and how the models are tested. Please consider adding this information to both the main text and appendix.

2. The results in the tables lack error bars.

3. The frequency analysis is limited to a simple visualisation of the perturbations’ spectrum. Could you provide a more quantitative comparison? Additionally, please define the spectrum and add a scale to the spectrum plots (ticks are missing).

**Questions:**

1. Can you insert a frequency constraint on the generated perturbations by modifying the loss function? What happens if you impose high or low-frequency perturbations when using different combinations of attack methods? Could this be a way to test the requirement for transferable perturbations to be high-frequency?

2. In some cases, the proposed algorithm ‘ours’ performs better than the one designed for a given learning paradigm. See for example Table 1, SimSiam and SimCLR: ‘ours’ outperforms ‘CP’. Is there an intuitive explanation for this? Analogously, in Table 6, some results indicate that transferred attacks work better on architectures they were not originally designed for. Error bars would be particularly useful in these cases.

3. In section 4.3 you say that MoCov2 is used both for poison generation and evaluation. Could you please elaborate? Also, what does Framework mean in Table 4?

4. I am not sure I understand the result in Fig. 3, panel C. Is there an explanation for the model performing better when trained on perturbed data? Why only in this case?

5. What is the computational cost of the proposed strategies? A comment on the computational costs should be added to the main text.

Comments:

1. Captions could be expanded to include experimental details (what model is attacked, using which dataset, etc) and explain the acronyms.

2. The tables show results for supervised, semi-supervised, and unsupervised algorithms. Shouldn’t SimSiam and SimCLR be grouped together in the tables?

3. I would suggest using an acronym for the proposed method instead of calling it ‘ours’.

4. It would be interesting to have the perturbed images displayed alongside their clean versions.

---

> ### Author Response · Authors · 2023-11-20
>
> 1. What is the motivation of this paper?
>
> We explain the motivation of our work in the introduction. To put it simple, since the three main learning paradigms (supervised, semi-supervised and unsupervised learning) can reach similar performance in various machine learning tasks, the victim may choose any learning algorithm from them to train their model. However, the current availability poisoning attacks often target for a specific learning paradigm and have poor transferability across different learning paradigms. Therefore, we propose to study the attack transferability on the level of learning paradigms.
>
> 2. More analysis on frequency and insert a loss function to modify frequency?
>
> Please refer to our answer to Q2 in global response. The current visualizations can clearly show that TAP and CP can generate poisons with similar high-frequency patterns, which are more different to the low-frequency poisons generated by EM and both low- and high-frequency poisons generated by TUE. Since we do not claim that only high-frequency poisons can have transferability nor poisons with higher frequency can have better transferability, we think that using a loss function to modify the frequency or providing quantitative results will not provide further explanations.
>
> 3. Results with error bars?
>
> We admit that using error bars will make the results more reliable. We will provide it later due to the limited computational resources.
>
> 4. What does framework mean and elaborate the use of MoCov2 in both poison generation and evaluation?
>
> The Framework means the working mechanisms of different unsupervised contrastive learning, such as SimCLR and MoCov2. So in section 4.3, we change the framework from SimCLR to MoCov2 to generate the poisons and also involve MoCov2 as one unsupervised contrastive learning algorithm to evaluate the generated poisons.
>
> 5. Question about figure 3?
>
> As for panel C in figure 3 where the training algorithm is semi-supervised learning, both the labeled and unlabeled data are partially poisoned, so for the model trained with poisoned data, the algorithm is more likely to involve the unlabeled data that will be helpful for the training including part of the unlabeled poisoned data, which means the model can learn more information and the results will be better. While for other settings, all poisoned data take part in the training originally, so the results will be worse than model trained with the rest of the clean data.
>
> 6. Computational cost?
>
> Please refer to the new Table 10 in the updated paper. As our method involves contrastive learning, the optimization time is comparatively longer, but it is still less than CP.

---

> > ### Comment · Reviewer_s6uj · 2023-11-22
> >
> > Dear authors,
> >
> > Thank you for the effort put into the rebuttal and for providing the new experimental results. As I mentioned in my review, this work tackles an important topic - transferability across learning paradigms - and it provides an effective method by combining existing high-frequency poisoning techniques. While I acknowledge the relevance of the contribution, I believe that there is room for improvement. This should involve providing more evaluations and estimating confidence intervals for the results, investigating the high-frequency hypothesis, offering more context and explanation of the motivation, and adding more details about the learning algorithms utilized and the employed poisoning methods. It will be important to address these points in the final version of the manuscript.

---

### Author Response · Authors · 2023-11-20

We thank all the reviewers for their valuable reviews.

We conduct additional experiments and provide the results in the updated paper. Specifically, we evaluate the baseline methods on CIFAR-100 (table 8) which are required by reviewer uUSA and f8DL, we apply JPEG operation (proposed by Liu et al.[6]) to defend against our attacking method (table 9) which are required by reviewer TTnu and uUSA, we calculate the computational cost for each attacking method (table 10) which are required by reviewer s6uj, miSr and f8DL, and we change the framework of baseline methods CP and TUE (table 11) which are required by reviewer f8DL.

Q1. The setting that poisoning 100% training data is not practical?

For the attack setting, we follow the previous works[1-4] where the attacker is assumed to have the capability to poison the whole training dataset with small norm-bounded perturbations. We admit that this assumption is strong in practice and it would be more meaningful to consider the partial poisoning setting where the attacker can not modify the whole training dataset. However, studying the full-poisoning scenario provides a worst-case analysis about the learning system against poisoning attacks, which enables the victim to understand which learning method is most resilient against small training-time perturbations. In addition, we also note that even under the strongest full-poisoning setting, all the aforementioned prior attacks have poor success rate if the victim makes use of different learning paradigms to train the model. We think it’s meaningful to explore the transferability of availability poisoning attacks across different learning paradigms with this strong assumption first, and then extend it to more constrained settings, which we plan to study in our future work.

Q2. The poisoning method is not novel?

Our method utilizes the characteristics of poisons from both supervised and unsupervised contrastive learning to enable better transferability, though simple yet effective. We also provide other alternatives to show that our method is not just a combination of previous works, e.g., two naive ways of combination (HF and CC) and another two variants of our method (SCP and WTP). The transferability of all these alternatives are much worse than our method.

Q3. Why TP can generate poisoning perturbations with high frequency and why high-frequency poisons have better transferability?

Based on the visualizations of the poisons generated by previous works, we find that both TAP[2] and CP[3] can produce poisons with high frequency. Since our attack is mainly based on these two methods, the poisoned data generated by our method are expected to have the similar high-frequency characteristics as well. We do not claim that only high-frequency poisons can have transferability nor poisons with higher frequency can have better transferability, but instead, we just utilize the similar high-frequency patterns and reduce the conflicts from different poisons, which is also supported by the results from other alternative methods. It is unclear which way of optimization can produce poisons with high or low frequency, which is not explained theoretically in the original papers[1,2,3] either. We admit that the current findings are based on visualizations and empirical results. We plan to conduct more theoretical analysis in future work.

[1] Unlearnable Examples: Making Personal Data Unexploitable. Huang et al., ICLR, 2021.

[2] Adversarial Examples Make Strong Poisons. Fowl et al., NeurIPS, 2021.

[3] Indiscriminate Poisoning Attacks on Unsupervised Contrastive Learning. He et al., ICLR, 2023.

[4] Transferable Unlearnable Examples. Ren et al., ICLR, 2023.

[5] Supervised Contrastive Learning. Khosla et al., NeurIPS, 2020.

[6] Image Shortcut Squeezing: Countering Perturbative Availability Poisons with Compression. Liu et al., ICML, 2023.

---

### Meta-Review · Area_Chair_gPsP · 2023-12-05

**Metareview:**

This paper finds that existing availability attacks are brittle and fail when the victim doesn’t train the same way as anticipated by the adversary.  The authors propose to solve this problem with a more transferable availability attack that transfers to victims regardless of their learning strategy.  Reviewers pointed out issues concerning novelty, lack of defenses, and scalability.  One reviewer pointed out a possible bug in the code which was not ultimately rebutted by the authors.  Some reviewers also complained that the method requires poisoning all training samples, but I am not bothered by this because it’s a well-known and well-motivated threat model which seemingly some reviewers are not familiar with, so I am not factoring this point into my decision.  Nonetheless, I am inclined to reject the paper based on the previous reasons.

**Justification For Why Not Higher Score:**

The evaluations are currently missing defenses aside from JPEG, which was pointed out by the reviewers.  The work also has very limited novelty and is not scalable.  One reviewer pointed out a possible bug in the code which was not ultimately rebutted by the authors.

**Justification For Why Not Lower Score:**

N/A

---

### Decision · Program_Chairs · 2024-01-16

Reject